# Pulmonary Artery Pulsatility Index and Hemolysis during Impella-Incorporated Mechanical Circulatory Support

**DOI:** 10.3390/jcm11051206

**Published:** 2022-02-23

**Authors:** Makiko Nakamura, Teruhiko Imamura, Yuki Hida, Koichiro Kinugawa

**Affiliations:** Second Department of Internal Medicine, University of Toyama, 2630 Sugitani, Toyama 930-0194, Japan; nakamura@med.u-toyama.ac.jp (M.N.); hao_oo7@yahoo.co.jp (Y.H.); kinugawa@med.u-toyama.ac.jp (K.K.)

**Keywords:** ventricular assist device, hemodynamics, right ventricular failure, extracorporeal membrane oxygenation

## Abstract

Background: Impella is a percutaneous transcatheter left ventricular assist device. Device-related hemolysis is a serious complication that is sometimes encountered depending on the device position, device speed, and support duration. However, the impact of hemodynamics on the occurrence of hemolysis remains unknown. In this study, we aimed to clarify the relationships between hemodynamics, especially right ventricular function, and the occurrence of hemolysis during Impella-incorporated mechanical circulatory support. Methods: Consecutive patients who received Impella (2.5, CP, and 5.0) support at our institute between March 2018 and July 2021 were retrospectively included. The relationships between the pulmonary artery pulsatility index (PAPi) immediately after Impella insertion and the occurrence of hemolysis were investigated. Results: Forty-two patients (median 71 years old, 60% men) were included. Hemolysis occurred in 20 patients (48%). A cutoff of PAPi to predict hemolysis was calculated as 1.3, with 80.0% sensitivity and 72.7% specificity. Lower PAPi (<1.3) significantly correlated with the occurrence of hemolysis with an odds ratio of 11.65 (95% confidence interval 1.58–85.98, *p* = 0.017), adjusted for other potential confounders. Survival discharge was significantly lower in patients with lower PAPi (<1.3) (50% vs. 86%, *p* = 0.019). Conclusions: The results of this study suggest that patients with right ventricular impairment indicated by lower PAPi following the initiation of Impella-incorporated mechanical circulatory support have a higher risk of hemolysis.

## 1. Introduction

The use of the percutaneous left ventricular assist device (LVAD) Impella (Abiomed, Danvers, MA, USA) is increasing in the setting of cardiogenic shock with acute myocardial infarction or acute decompensated heart failure. However, device-related hemolysis remains a serious complication [1].

Free plasma hemoglobin, which is increased by active hemolysis, triggers tissue hypoxia and cell death. Free plasma hemoglobin scavenges nitric oxide, leading to inappropriate vasoconstriction, endothelial dysfunction, and platelet aggregation. Consequently, hemolysis may cause transfusion-requiring anemia and acute kidney injury. Severe hemolysis sometimes requires inappropriate early device removal during unstable hemodynamics [2,3].

The causes of hemolysis during Impella support are multifactorial, and include inappropriate device positioning, incremental device speed, and prolonged support duration [2]. For the prevention of hemolysis, echocardiography-guided device position adjustment, reduced rotation speed, and early device explantation are encouraged [2,4].

However, the impact of hemodynamic status on the occurrence of hemolysis remains uncertain. During durable LVAD support, sub-clinical right ventricular failure is associated with pump thrombosis [5]. Additionally, a relationship between suspected Impella-pump thrombosis and hemolysis was recently reported [6]. Given together, right ventricular failure might be associated with device thrombosis and hemolysis during Impella support. In this study, we investigated the relationships between hemodynamics, particularly right ventricular function parameters, and the occurrence of hemolysis during Impella-incorporated mechanical circulatory support.

## 2. Methods

### 2.1. Patient Selection

Initially, 48 consecutive patients who received Impella support between March 2018 and July 2021 at our institute were retrospectively investigated. We excluded six patients: four patients with life-threatening bleeding refractory to hemostat and two patients with missing data. Of those four patients, one patient had chest bleeding due to trauma associated with cardiopulmonary resuscitation and the other three patients had surgical bleeding; these patients eventually died due to blood loss, despite a low dose of heparin purge solution. Finally, 42 patients were included. The present study was approved by the local Institutional Ethical Review Board and all patients gave their informed consent before this study was conducted.

### 2.2. Device Selection

Impella was inserted via the femoral or axillary artery. The device type was selected according to the diameters of the access vessels and the underlying disease, indications, and hemodynamics. All these decisions were made by discussion among the multidisciplinary team.

For example, Impella 2.5/CP was inserted before primary percutaneous coronary intervention (PCI) in ST elevation myocardial infarction. We inserted Impella 2.5 until October 2019 and CP after November 2019, due to the issue of approved timing. If end-organ dysfunction persisted or progressed even after primary PCI, veno-arterial extracorporeal membrane oxygenation (ECMO) was used concomitantly with Impella for the end-organ recovery.

In case of INTERMACS Profile 1 (i.e., refractory ventricular fibrillation, cardiac arrest on cardiopulmonary arrest resuscitation, or severe cardiogenic shock accompanied by hepatorenal dysfunction), we first inserted veno-arterial ECMO and performed PCI if needed. We inserted Impella 2.5 or CP in addition to veno-arterial ECMO in case of left ventricular dysfunction, elevated left ventricular end-diastolic pressure, or both.

In case of INTERMACS Profile 2 (i.e., acute decompensated heart failure of cardiomyopathy with progressive end-organ dysfunction against inotropes) we inserted Impella 5.0 for a bridge to recovery or durable LVAD.

### 2.3. Management of Impella Support

A pulmonary artery catheter was inserted for the continuous monitoring of hemodynamics before or immediately after Impella insertion. The device position was adjusted properly and repeatedly by echocardiography.

All patients received anticoagulation therapy with a targeted activated whole clotting time between 160 and 180 s, and the dose of heparin purge solution as well as systemic heparin administration were adjusted accordingly.

### 2.4. Clinical Data

A primary endpoint was the incidence of hemolysis, which was diagnosed by the existence of hemoglobinuria and elevated levels of lactate dehydrogenase >2.5-fold of the upper normal range. Clinical data were retrieved, including laboratory data on admission and during Impella support and echocardiographic data on admission.

Hemodynamic data, including central venous pressure and the pulmonary artery pulsatility index (PAPi), were measured immediately after the initiation of Impella support (major independent variables in the present study). Activated whole-blood clotting time and activated partial thromoboplastin time obtained during Impella support were all averaged.

### 2.5. Management of Hemolysis

If the patients had hemolysis, device position and the anticoagulation level were immediately investigated. If hemolysis persisted against the correct device position, the anticoagulation was strengthened and the device support levels were decreased. Hemodynamics were measured and adjusted by fluid infusion, administration of intravenous inotropes, or both in order to support the decreased support flow. Haptoglobin was also administered to reduce hemolysis-induced renal impairment. Device upgrade or withdrawal were considered for severe hemolysis refractory to these procedures.

### 2.6. Statistical Analysis

Statistical analyses were performed using JMP pro ver14.0 (SAS Institute Japan Ltd., Tokyo, Japan). Variables with *p* < 0.05 were considered significant. Continuous data were described as medians and interquartile ranges and compared between two groups using the Mann–Whitney U test. Categorical data were compared between two groups using the chi-squared test or Fisher’s exact test where appropriate.

A cut-off of PAPi for hemolytic events was calculated using a receiver operating characteristic analysis and the cohort was stratified into two groups: a lower PAPi group and higher PAPi group. Logistic regression analyses were performed to investigate the relationship between baseline characteristics, including PAPi, and the occurrence of hemolysis. Variables with *p* < 0.05 in the univariable analysis were included in the multivariable analysis.

Hemolysis during ECMO support is observed at an incidence between 5% and 18% [3], and we included the use of ECMO in the multivariable analysis irrespective of the statistical significance in the univariable analysis.

Clinical outcomes other than hemolysis were compared between the dichotomized PAPi groups. The 150-day survivals were also compared by Kaplan–Meier analyses and the log-rank test.

## 3. Results

### 3.1. Baseline Characteristics

Forty-two patients who received Impella (2.5, CP, and 5.0) support were included. Baseline demographic data are summarized in Table 1. The median age was 71 years, and 25 (60%) patients were male. Twenty-seven (64%) patients had acute myocardial infarction and concomitantly underwent PCI. A total of ten patients received concomitant ECMO support: eight of them had acute myocardial infarction, one had fulminant myocarditis, and the other had acute decompensated heart failure due to non-ischemic cardiomyopathy assigned to INTERMACS Profile 1.

On admission, the median plasma level of B-type natriuretic peptide was 704 (350–1502) pg/mL and the median left ventricular ejection fraction was 29% (20–38%) (Table 2).

### 3.2. Clinical Parameters during Impella Support

Immediately after Impella support, median arterial pressure was 75.5 (70.0–85.3) mmHg and PAPi was 1.30 (0.72–2.31). During device support for a median of 6 days, the average activated whole-blood clotting time was 162 (149–171) s and the maximum lactate dehydrogenase level was 1138 (691–2124) IU/L (Table 3).

### 3.3. Impact of PAPi on Hemolysis

During Impella support, a total of 20 patients (48%) encountered hemolysis events; 11 patients (52%) in the Impella 2.5 group, 7 patients (63%) in the Impella CP group, and 2 patients (20%) in the Impella 5.0 group. Their incidences were not significantly different between the devices. A cut-off of PAPi to predict hemolysis was calculated as 1.3 with an area under the curve of 0.723 (80.0% sensitivity and 72.7% specificity; Figure 1). The incidence of hemolysis was significantly higher in the lower PAPi group (<1.3) than in the higher PAPi group (≥1.3) (70% vs. 27%, *p* = 0.0124; Figure 2).

Lower PAPi (<1.3) was identified as a significant risk factor for hemolysis with an unadjusted odds ratio of 6.222 (95% confidence interval 1.630–23.758, *p* = 0.0049) and an odds ratio of 11.654 (95% confidence interval 1.580–85.982, *p* = 0.017) adjusted for serum creatinine kinase levels (>150 IU/L), which was also significant in the univariable analyses, and concomitant use of ECMO, which carries a potential risk of hemolysis [7] but did not reach statistical significance in the univariable analysis (Table 4). We did not include the occurrence of acute myocardial infarction in the multivariable analysis, considering its multicollinearity with higher creatinine kinase.

We performed a sub-analysis for those without ECMO supports (*n*= 32). Lower PAPi (<1.3) was again identified as a significant risk factor for hemolysis, with an unadjusted odds ratio of 10.667 (95% confidence interval 1.743–65.270, *p* = 0.010) and an odds ratio of 6.282 (95% confidence interval 0.014–43.184, *p* = 0.062) adjusted for acute myocardial infarction, which was also significant in the univariable analyses (Table 5).

### 3.4. Potential Trigger of Hemolysis

Among 20 hemolytic events, 10 events occurred following the unique triggers: 7 occurred following the intravenous administration of furosemide and 3 occurred after massive bleeding from the access site. Of note, seven of ten (70%) had lower PAPi (<1.3). In addition, one of them had accompanied pump thrombosis as well as hemolysis after massive bleeding.

### 3.5. Device Upgrade to Manage Hemolysis

One patient with ventricular septum rupture, with PAPi <1.3, experienced device upgrade from Impella 2.5 to Impella 5.0 to manage hemolysis and maintain systemic perfusion.

### 3.6. Other Clinical Outcomes

The prevalence of clinical outcomes other than hemolysis were stratified by PAPi (Table 6). There was a significantly higher rate of Impella upgrade and/or concomitant use of ECMO in the lower PAPi group. Impella was removed following hemodynamic recovery in 33 (79%) patients, while 6 (14%) were bridged to durable LVAD implantation.

Survival discharge rates were significantly lower in the lower PAPi group (50% vs. 86%, *p* = 0.0186, Table 6). The 30-day survival rates were not stratified by PAPi (*p* > 0.05), whereas the 150-day survival was significantly lower in the lower PAPi group (42% vs. 81%, *p* = 0.0451, Figure 3).

### 3.7. Stratification of Patient Characteristics by PAPi

Patients with lower PAPi received higher support flow with concomitant ECMO as well as mechanical ventilation (Table 1). On admission, these patients had higher lactate levels with normal left ventricle size (Table 2). During Impella support, maximum lactate dehydrogenase and maximum creatinine kinase levels were higher in the lower PAPi group, whereas the average activated whole-blood clotting time did not significantly differ, irrespective of PAPi (Table 3).

## 4. Discussion

In the present study, we investigated the relationship between PAPi and the occurrence of hemolysis during Impella-incorporated mechanical circulatory support. A lower PAPi (<1.3) measured immediately after the initiation of Impella support independently correlated with the occurrence of hemolysis.

### 4.1. Multiple Causes of Hemolysis during Impella Support

Device-related hemolysis is one of the critical complications encountered during all mechanical circulatory supports including Impella. Device-related hemolysis is in summary caused by shear stress along the impeller blades, which is affected by various parameters including device speed, device support duration, mechanical obstruction (inlet part and outlet part), and device lumen size [2]. Concomitant ECMO use might also increase the risk of hemolysis [7].

Interventions to some of these risk factors are attempted to prevent or manage hemolysis, including the adjustment of device speed, early device removal, echo-guided device position manipulation, volume loading to prevent device suction, management of device thrombosis, and upgrade of device size [2].

In this study, concomitant ECMO use was not associated with hemolysis. Acute hemodynamics deterioration indicated by incremental serum creatinine kinase level was associated with hemolytic events, probably due to unstable hemodynamics and concomitant risk factors described above.

### 4.2. PAPi and Hemolysis

In the previous study, baseline lower PAPi (<0.96) was associated with right ventricular failure following durable LVAD implantation [8]. Thus, baseline lower PAPi is considered to indicate sub-clinical right ventricular failure, which might worsen and clinically become actualized due to LVAD-induced incremental venous return. Furthermore, sub-clinical right ventricular failure during durable LVAD support is associated with hemocompatibility-related adverse events including pump thrombosis [5].

In addition to the above-described risk factors of hemolysis, we would like to propose a recently introduced hemodynamic parameter: PAPi [9]. A decrease in preload on the left ventricle due to impaired right ventricular function, indicated by lower PAPi, would reduce intra-device blood flow in the left ventricle and might increase shear stress and facilitate hemolysis.

### 4.3. Clinical Implications

We recommend the following strategies in cases of lower PAPi (<1.3) measured immediately after insertion of Impella: (1) be aware of the risk of hemolysis and repeatedly check the device position as well as anticoagulation level; (2) prevent hypovolemic status and avoid unnecessary diuretics use to prevent a further decrease in preload on the left ventricle; and (3) consider device upgrade if hemolysis is refractory.

### 4.4. Limitations

This was a retrospective study conducted at a single center using a small cohort. We included ten ECMO patients. Veno-arterial ECMO may affect PAPi, central venous pressure, and other hemodynamic parameters. However, ECMO use was not significantly associated with the primary endpoint. We performed a sub-analysis among those without ECMO supports, and the findings were similar to those of the overall cohort. In real-world practice, it is often challenging to manage those with unstable hemodynamics using Impella alone, and other mechanical circulatory supports including ECMO are reliable collaborative tools. This is a rationale for our inclusion of those with ECMO supports in addition to those with Impella alone.

We did not measure plasma free hemoglobin levels, which are one of the markers of hemolysis, because this was not covered by insurance and cannot be measured commercially.

We performed multivariable analyses using only three variables due to the relatively small cohort size and event numbers. Other uninvestigated confounders may have existed. We did not assess right ventricular function using echocardiography. Accurate assessment of right ventricular function by echocardiography is often challenging due to the existence of multiple mechanical devices. We did not include data for right atrial pressure or pulmonary artery wedge pressure due to the large amount of missing data. In addition, right ventricular function might change following Impella insertion [10], and the incidence might vary depending on device type.

Although multiple studies have investigated PAPi, it has yet to be adequately explored. Further investigation is needed to clarify the association between the absolute value of PAPi and right ventricular function in patients with acute myocardial infarction and/or following LVAD implantation.

## 5. Conclusions

Lower PAPi measured immediately after insertion of Impella correlated with the occurrence of hemolysis during Impella-incorporated mechanical circulatory support. Careful observation and aggressive interventions to prevent hemolysis would be required for such a high-risk cohort. Clinical implication of PAPi-guided Impella management to prevent hemolysis requires further validation study.

## Figures and Tables

**Figure 1 jcm-11-01206-f001:**
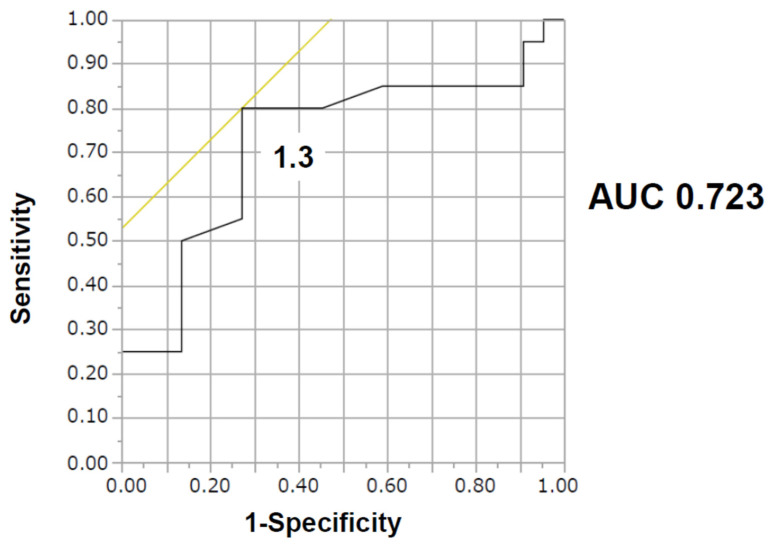
The cut-off pulmonary artery pulsatility index for hemolysis was calculated as 1.3 with an area under the curve of 0.7361, by receiver operating characteristic analyses. AUC, area under the curve.

**Figure 2 jcm-11-01206-f002:**
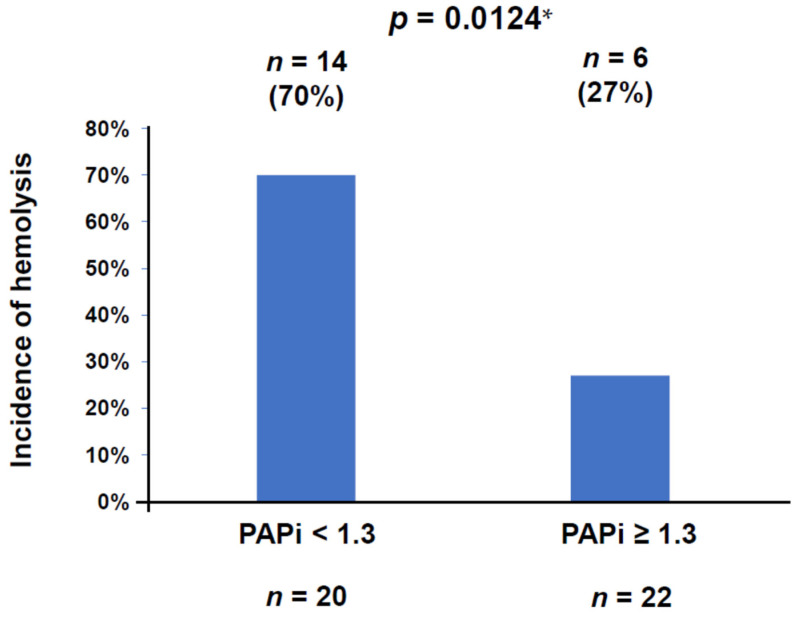
The incidence of hemolysis was significantly higher in the lower PAPi (<1.3) group than in the higher PAPi (≥1.3) group (70% vs. 27%, *p* = 0.0124) according to Fisher’s exact test. PAPi, pulmonary artery pulsatility index. * *p* < 0.05 by Fisher’s exact test

**Figure 3 jcm-11-01206-f003:**
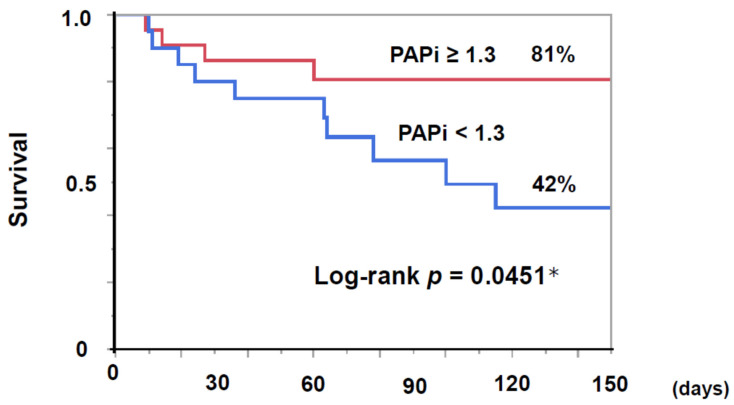
Survival in the lower PAPi (<1.3) group was significantly lower than the higher PAPi (≥1.3) group (42.3% vs. 80.6%, *p* = 0.0451) by the Kaplan–Meier analyses and log-rank test. PAPi, pulmonary artery pulsatility index. * *p* < 0.05 by log-rank test.

**Table 1 jcm-11-01206-t001:** Baseline demographic characteristics.

	Total (*n* = 42)	PAPi < 1.3 (*n* = 20)	PAPi ≥ 1.3 (*n* = 22)	*p*-Value
Age	71 (58, 82)	71.5 (57.3, 75)	71 (61.5, 82.5)	0.251
Female gender	17 (40%)	7 (35%)	10 (45%)	0.5430
Body surface area (m^2^)	1.60 (1.43, 1.69)	1.66 (1.56, 1.77)	1.55 (1.41, 1.63)	0.0101 *
Body mass index (kg/m^2^)	21.4 (19.4, 22.6)	22.4 (21.4, 23.9)	19.7 (18.3, 22.1)	0.0026 *
Acute myocardial infarction	27 (64%)	16 (80%)	11 (50%)	0.0577
Atrial fibrillation	13 (31%)	7 (35%)	6 (27%)	0.741
Old brain infarction	10 (24%)	4 (20%)	6 (27%)	0.7225
Impella 2.5	21 (50%)	9 (45%)	12 (55%)	0.7579
Impella 5.0	12 (29%)	4 (20%)	8 (36%)	0.3148
Impella CP	11 (26%)	9 (45%)	2 (9%)	0.0132 *
Extracorporeal membrane oxygenation	10 (24%)	10 (50%)	0 (0%)	0.0001 *
Via femoral artery insertion	32 (76%)	17 (85%)	15 (68%)	0.2842
Antiplatelet therapy	25 (60%)	14 (70%)	11 (50%)	0.2225
Support duration (days)	6 (3, 9)	6 (3.3, 9)	6 (3, 10)	0.8394
Continuous hemodiafiltration	11 (26%)	5 (25%)	6 (27%)	1.0000
Mechanical ventilation	23 (55%)	16 (80%)	7 (32%)	0.0023 *

PAPi, pulmonary artery pulsatility index. * *p* < 0.05 by Mann–Whitney U test or Fisher’s exact test as appropriate.

**Table 2 jcm-11-01206-t002:** Laboratory and echocardiography data on admission.

	Total (*n* = 42)	PAPi < 1.3 (*n* = 20)	PAPi ≥ 1.3 (*n* = 22)	*p*-Value
Laboratory data				
Lactate (mmol/L)	3.0 (1.3, 6.75)	6.8 (2.9, 9.5)	1.4 (1.1, 3.1)	0.0001 *
Hemoglobin (g/dL)	12.1 (10.1, 14.2)	13.4 (10.6, 14.9)	11.3 (9.7, 13.7)	0.0778 *
Serum total bilirubin (mg/dL)	0.8 (0.5, 1.1)	0.6 (0.4, 1.0)	0.8 (0.6, 1.2)	0.2285
Serum creatinine (mg/dL)	1.21 (0.95, 1.60)	1.13 (0.93, 1.47)	1.38 (0.99, 2.05)	0.1275
Serum creatinine kinase (IU/L)	205 (69, 544)	288 (128, 3380)	78 (53, 308)	0.0060 *
Plasma B-type natriuretic peptide (pg/mL)	704 (350, 1502)	474 (76, 1103)	817 (491, 2261)	0.0161 *
N-terminal pro B-type natriuretic peptide (pg/mL)	5532 (2023, 13,944)	4736 (349, 6925)	9423 (4334, 21,567)	0.0014 *
Echocardiographic data				
Left ventricular end-diastolic diameter (mm)	50 (44, 65)	48 (41, 53)	56 (45, 64)	0.0123 *
Left ventricular ejection fraction (%)	29 (20, 38)	30 (20, 37)	27 (20, 37)	0.7985
Mitral valve regurgitation	1 (0, 2)	0 (0, 1)	1 (1, 2)	0.0131 *

PAPi, pulmonary artery pulsatility index. Mitral regurgitation was graded as 0 (none), 1 (mild), 2 (moderate), and 3 (severe). * *p* < 0.05 by Mann–Whitney U test or Fisher’s exact test as appropriate.

**Table 3 jcm-11-01206-t003:** Hemodynamic and laboratory data during Impella support.

	Total (*n* = 42)	PAPi < 1.3 (*n* = 20)	PAPi ≥ 1.3 (*n* = 22)	*p*-Value
Hemodynamic data just after Impella initiation				
Mean arterial pressure (mmHg)	75.5 (70.8, 85.3)	75.5 (70.3, 85.8)	75.8 (70.8, 82.0)	0.8897
Central venous pressure (cmH_2_O)	8 (5, 9)	9 (8, 10)	5.5 (4, 8)	<0.0001 *
Pulmonary artery pulsatility index	1.30 (0.74, 2.31)	0.69 (0.34, 1.00)	2.25 (1.58, 3.13)	<0.0001 *
Laboratory data during Impella support				
Maximum lactate dehydrogenase on support (IU/L)	1138 (691, 2124)	2004 (1198, 3560)	783 (398, 1283)	0.0008 *
Maximum total bilirubin on support (mg/dL)	1.9 (1.2, 2.5)	2.1 (1.5, 3.3)	1.9 (1.2, 2.0)	0.2113
Maximum creatinine on support (mg/dL)	1.53 (1.00, 2.29)	1.53 (1.05, 2.19)	1.48 (0.96, 2.71)	0.9699
Maximum creatinine kinase on support (IU/L)	1250 (198, 6195)	6039 (2415, 12738)	238 (120, 991)	<0.0001 *
Averaged activated whole blood clotting time (s)	162 (149, 171)	160 (134, 173)	164 (152, 171)	0.2779
Averaged activated partial thromboplastin time (s)	50.6 (37.1, 64.4)	50.0 (35.0, 63.4)	50.6 (41.3, 65.6)	0.4119
Averaged activated whole blood clotting time < 160 (s)	15 (36%)	9 (45%)	6 (27%)	0.3357

PAPi, pulmonary artery pulsatility index. * *p* < 0.05 by Mann–Whitney U test or Fisher’s exact test as appropriate.

**Table 4 jcm-11-01206-t004:** Univariable and multivariable analyses for the hemolytic event.

	Univariable Analyses		Multivariable Analyses	
	Odds Ratio (95% CI)	*p*-Value	Odds Ratio (95% CI)	*p*-Value
Age (years)	1.004 (0.960–1.050)	0.849		
Female gender	0.646 (0.186–2.243)	0.492		
Acute myocardial infarction	4.000 (1.008–15.868)	0.049 *		
Impella 2.5	1.467 (0.434–4.951)	0.537		
Impella CP	2.423 (0.585–10.030)	0.214		
Impella 5.0	0.438 (0.108–1.771)	0.247		
Extracorporeal membrane oxygenation	1.929 (0.455–8.182)	0.373	0.227 (0.026–2.279)	0.208
Via femoral artery insertion	2.644 (0.578–12.095)	0.210		
Creatinine kinase (IU/L)	1.0004 (0.9999–1.0010)	0.032 *		
Creatinine kinase >150 (IU/L)	6.429 (1.662–24.860)	0.004 *	6.622 (1.410–31.096)	0.017 *
Activated whole-blood clotting time (s)	0.992 (0.963–1.022)	0.602		
Pulmonary artery pulsatility index	0.695 (0.429–1.126)	0.091		
Pulmonary artery pulsatility index <1.3	6.222 (1.630–23.757)	0.0049 *	11.654 (1.580–85.982)	0.017 *
Central venous pressure (cmH_2_O)	1.212 (0.961–1.530)	0.050		

* *p* < 0.05 by logistic regression analysis. CI, confidence interval.

**Table 5 jcm-11-01206-t005:** Univariable and multivariable analyses for the hemolytic event among only Impella use patients.

	Univariable Analyses		Multivariable Analyses	
	Odds Ratio (95% CI)	*p*-Value	Odds Ratio (95% CI)	*p*-Value
Age (years)	1.022 (0.967–1.080)	0.434		
Female gender	0.400 (0.091–1.762)	0.217		
Acute myocardial infarction	9.429 (1.603–55.447)	0.013 *	5.578 (0.840–37.028)	0.075
Impella 2.5	1.667 (0.407–6.818)	0.475		
Impella CP	4.444 (0.712–27.759)	0.094		
Impella 5.0	0.208 (0.036–1.214)	0.060		
Via femoral artery insertion	23.818 (0.649–22.453)	0.115		
Creatinine kinase on admission (IU/L)	1.000 (1.000–1.001)	0.221		
Activated whole-blood clotting time (s)	0.964 (0.925–1.005)	0.065		
Pulmonary artery pulsatility index	0.725 (0.424–1.240)	0.184		
Pulmonary artery pulsatility index < 1.3	10.667 (1.743–65.270)	0.010 *	6.282 (0.014–43.184)	0.062
Central venous pressure (cmH_2_O)	1.087 (0.811–1.456)	0.575		

* *p* < 0.05 by logistic regression analysis. CI, confidence interval.

**Table 6 jcm-11-01206-t006:** Clinical outcomes.

	Total (*n* = 42)	PAPi < 1.3 (*n* = 20)	PAPi ≥ 1.3 (*n* = 22)	*p*-Value
Impella upgrade/concomitant use of ECMO	11 (26%)	11 (55%)	0 (0%)	<0.0001 *
Suction events	7 (17%)	5 (25%)	2 (9.1%)	0.2289
Pump thrombosis	1 (2.4%)	1 (5.0%)	0 (0%)	0.4762
Hemolysis	20 (48%)	14 (70%)	6 (27%)	0.0124 *
Recovery without mechanical circulatory support	33 (79%)	17 (85%)	16 (73%)	0.4595
Durable left ventricular assist device implantation	6 (14%)	1 (5.0%)	5 (23%)	0.187
30-day survival	34 (81%)	15 (75%)	19 (86%)	0.4454
Survival discharge	29 (69%)	10 (50%)	19 (86%)	0.0186 *

ECMO, extracorporeal membrane oxygenation. * *p* < 0.05 by Mann–Whitney U test or Fisher’s exact test as appropriate.

## Data Availability

Data are available by the corresponding author upon reasonable requests.

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
