# Peer review of "Pulmonary Artery Pulsatility Index and Hemolysis during Impella-Incorporated Mechanical Circulatory Support"

_jcm, 2022, doi:10.3390/jcm11051206_

Round 1
Reviewer 1 Report
The manuscript addresses the relationship between hemodynamics, right ventricular function parameters, and the occurrence of hemolysis during Impellla support. The study concludes patient that develop lower PAPI have increased risk of hemolysis. I have several comments
1)In the Device selection section the author mentions Impella insertion - please specify which Impella device is being referred to
2)I would not include patients who have both Impella and ECMO devise at the same time as this would first, increase the risk of hemolysis, second it would confound the data
3)In the method section please include the total number of patients that received an Impella and the total number of patients excluded.
4)Why wasn't plasma free hemoglobin measured. LDH is not the best marker of hemolysis
5)The groups should be further divided into Impella types. The hemolysis risk varies by device. Furthermore, the risk of developing RV dysfunction varies by device.
Author Response
Reviewer #1
General comment
The manuscript addresses the relationship between hemodynamics, right ventricular function parameters, and the occurrence of hemolysis during Impellla support. The study concludes patient that develop lower PAPI have increased risk of hemolysis. I have several comments
Response
We sincerely express our great appreciation for the reviewer’s recommendation to our manuscript. According to the reviewer’s comment, we revised our manuscript. Please read through our revised manuscript.
Comment 1
In the Device selection section the author mentions Impella insertion - please specify which Impella device is being referred to.
Response 1
Thank you for the reviewer’s important comment. Impella 2.5 and 5.0 had been approved for insurance since September 2017, and Impella CP had been commercially available since October 2019 in our country.
For that reason, we inserted Impella 2.5 and perform primary PCI in STEMI until October 2019, but inserted Impella CP after November 2019. If end-organ dysfunction still remained or progressed even after primary PCI, we added V-A CMO to maintain systemic perfusion.
In case of INERMACS Profile 1, i.e. refractory ventricular fibrillation, cardiac arrest on CRP, severe cardiogenic shock accompanied hepatorenal dysfunction such as SCAI-Shock stage E, we inserted V-A ECMO at first and perform CAG and PCI if needed. Because LV unloading with Impella during ECMO was associated with improved survival (Rev Cardiovasc Med. 2021 22;22(4):1503-1511), we added Impella 2.5 or CP for LV unloading in case of LV dysfunction and/or elevated LVEDP.
In case of INTERMACS Profile 2, i.e. acute decompensated heart failure of cardiomyopathy, we inserted Impella 5.0 for bridge to recovery or durable LVAD. We revised the description concerning device selection.
Before 1
Methods
Device selection:
The device type was selected according to the diameters of the access vessels and required support flow/duration. Veno-arterial extracorporeal membrane oxygenation (ECMO) was used concomitantly with Impella if support flow was not sufficient for end-organ recovery.
After 1
Methods
Device selection:
Impella was inserted via femoral or axillary artery. The device type was selected according to the diameters of the access vessels and the underlying disease, indications, and hemodynamics. All these decisions were made by the multidisciplinary team discussion.
For example, Impella 2.5/CP was inserted before primary percutaneous coronary intervention (PCI) in ST elevation myocardial infarction. We inserted Impella 2.5 until October 2019 and CP after November 2019, due to the issue of approved timing. If end-organ dysfunction persisted or rather progressed even after primary PCI, veno-arterial extracorporeal membrane oxygenation (ECMO) was used concomitantly with Impella for the end-organ recovery.
In case of INERMACS Profile 1, i.e. refractory ventricular fibrillation, cardiac arrest on cardiopulmonary arrest resuscitation, or severe cardiogenic shock accompanied by hepatorenal dysfunction, we inserted veno-arterial ECMO at first and performed PCI if needed. We inserted Impella 2.5 or CP in addition to veno-arterial ECMO in case of left ventricular dysfunction and/or elevated left ventricular end-diastolic pressure.
In case of INTERMACS Profile 2, i.e. acute decompensated heart failure of cardiomyopathy with progressive end-organ dysfunction against inotropes, we inserted Impella 5.0 for bridge to recovery or durable LVAD.
Comment 2
I would not include patients who have both Impella and ECMO devise at the same time as this would first, increase the risk of hemolysis, second it would confound the data
Response 2
We agree with the reviewer’s comment. We performed similar analyses among 32 patients without ECMO supports. The cut-off value of PAPi for hemolytic event was calculated as 1.3, which was the same value as that calculated among 42 total patients including ECMO support. Lower PAPi (< 1.3) was also identified as a significant risk factor for hemolysis with an unadjusted odds ratio of 10.667 (95% confidence interval 1.743–65.270, p = 0.010), and an odds ratio of 6.282 (95% confidence interval 0.014–43.184, p = 0.062) adjusted for acute myocardial infarction. We added these findings in the text.
Before 2
Results
None
Table
None
Discussion
Limitations:
None
After 2
Results
Impact of PAPi on hemolysis:
We performed a sub-analysis for those without ECMO supports (N = 32). Lower PAPi (< 1.3) was again identified as a significant risk factor for hemolysis with an unadjusted odds ratio of 10.667 (95% confidence interval 1.743–65.270, p = 0.010) and an odds ratio of 6.282 (95% confidence interval 0.014–43.184, p = 0.062) adjusted for acute myocardial infarction, which was also significant in the univariable analyses (Table 5).
Table
See updated Table 5
Discussion
Limitations:
We performed a sub-analysis among those without ECMO supports, and the findings were almost similar to those of overall cohort. In the real-world practice, it is often challenging to manage those with unstable hemodynamics using Impella alone, and other mechanical circulatory supports including ECMO are reliable collaborative tools. This is a rationale why we included those with ECMO supports in addition to those with Impella alone.
Comment 3
In the method section please include the total number of patients that received an Impella and the total number of patients excluded.
Response 3
Thank you for the reviewer’s important comment. We revised the manuscript as suggested by the reviewer.
Before 3
Methods
Patient selection:
Consecutive patients who received Impella support … eventually died due to blood loss.
After 3
Methods
Patient selection:
Consecutive 48 patients who received Impella support … due to blood loss. Finally, 42 patients were included.
Comment 4
Why wasn't plasma free hemoglobin measured. LDH is not the best marker of hemolysis
Response 4
We sincerely agree with the reviewer’s comment. LDH elevation is also caused by other causes such as myocardial infarction, and not the best marker of hemolysis. However, we cannot measure the plasma free hemoglobin by the insurance and the measurement of plasma free hemoglobin is commercially not available in our country. We strengthened the limitation section.
Before 4
Limitations
We did not obtain data on plasma free hemoglobin levels and did not investigate the relationship between the occurrence of hemolysis and Impella support flow.
After 4
Limitations
We did not measure plasma free hemoglobin levels, which was one of the markers of hemolysis, because it was not covered by the insurance and cannot be measured commercially.
Comment 5
The groups should be further divided into Impella types. The hemolysis risk varies by device. Furthermore, the risk of developing RV dysfunction varies by device.
Response 5
Thank you for the reviewer’s critical comment. As the reviewer’s comment, Impella 5.0 has been reported as less frequency of hemolysis (J Card Surg. 2020;35(12):3310-3316), although univariable analysis for the hemolytic event in our study shown in Table 4 revealed no significant difference between Impella devices. We also showed as below the occurrence of hemolysis separately for each Impella device. There was no significant difference for the occurrence of hemolysis between Impella devices (p = 0.112), although Impella 5.0 had tendency of low risk for hemolysis.
|
  |
Impella 2.5 |
Impella CP |
Imeplla 5.0 |
|
Hemolysis |
11 (52%) |
7 (63%) |
2 (20%) |
|
Non-Hemolysis |
10 (48%) |
4 (36%) |
8 (80%) |
|
Total |
21 (50%) |
11 (26%) |
10 (24%) |
The risk of developing RV dysfunction may vary by device type, although not investigated in detail thus far. There was a report that the impaired RV function caused by LMT-AMI cardiogenic shock, was partially restored during Impella CP support in adult pigs (Intensive Care Med Exp. 2020 12;8(1):41). On the other hand, the right ventricular failure following Impella insertion had also been reported (J Heart Lung Transplant. 2021 Nov15;S1053-2498(21)02585-7). Further investigation is needed to clarify the risk of developing RV dysfunction among Impella devices. It may be dependent on the underlying diseases. We added the description to the manuscript.
Before 5
Results
None
Limitation
None
After 5
Results
Impact of PAPi on hemolysis:
During the Impella support, a total of 20 patients (48%) encountered hemolysis events; 11 patients (52%) in Impella 2.5, 7 patients (63%) in Impella CP, and 2 patients (20%) in Impella 5.0, which was not significantly different of incidence among Impella devices.
Limitation
In addition, right ventricular function might change following Impella insertion (J Heart Lung Transplant. 2021 Nov15;S1053-2498(21)02585-7), and the incidence might vary depending on the device types.

Reviewer 2 Report
Authors did great job. This is important concept and needs to be investigated further with RCTs. Few minor corrections:
molysis sometimes requires inappropriately early device removal while on unstable he- 49
with life-threatening bleeding refractory to hemostate and two patients with missing data. 68
ics was measured and adjusted by fluid infusion and/or administration of intravenous 105
baseline demographic data are summarized in Table 1. The median age was 71 years old 132
averaged activated whole blood clotting time was 162 (149-171) sec and the maximum 151
coagulation level as optimal range, and maintain preload of Impella are recommended for 272
Author Response
Reviewer #2
General comment
Authors did great job. This is important concept and needs to be investigated further with RCTs. Few minor corrections:
Response
We sincerely express our great appreciation for your comments that have improved our manuscript. According to the reviewer’s comment, we revised our manuscript. Please read through our updated manuscript.
Comment 1
Hemolysis sometimes requires inappropriately early device removal while on unstable he- 49
Response 1
Thank you for the pointed out. We corrected the word.
Before 1
Introduction
Severe hemolysis sometimes requires inappropriately early device removal while on unstable hemodynamics [2, 3].
After 1
Introduction
Severe hemolysis sometimes requires inappropriate early device removal while on unstable hemodynamics [2, 3].
Comment 2
With life-threatening bleeding refractory to hemostate and two patients with missing data. 68
Response 2
We corrected the word.
Before 2
Methods
We excluded six patients: four patients with life-threatening bleeding refractory to hemostate and two patients with missing data.
After 2
Methods
We excluded six patients: four patients with life-threatening bleeding refractory to hemostat and two patients with missing data.
Comment 3
ics was measured and adjusted by fluid infusion and/or administration of intravenous 105
Response 3
Thank you for the comment. We revised the word.
Before 3
Methods
Management of hemolysis:
Hemodynamics was measured and adjusted by fluid infusion and/or administration of intravenous inotropes to support the decreased support flow.
After 3
Methods
Management of hemolysis:
Hemodynamics were measured and adjusted by fluid infusion and/or administration of intravenous inotropes to support the decreased support flow.
Comment 4
baseline demographic data are summarized in Table 1. The median age was 71 years old 132
Response 4
We corrected the word appropriately.
Before 4
Results
Baseline demographic data are summarized in Table 1.
After 4
Results
Baseline demographic data was summarized in Table 1.
Comment 5
averaged activated whole blood clotting time was 162 (149-171) sec and the maximum 151
Response 5
We corrected the word.
Before 5
Results
Clinical parameters during Impella support:
During device support for a median of 6 days, the averaged activated whole blood clotting time was 162 (149-171) sec and the maximum lactate dehydrogenase level was 1138 (691-2124) IU/L (Table 3).
After 5
Results
Clinical parameters during Impella support:
During device support for a median of 6 days, the average activated whole blood clotting time was 162 (149-171) sec and the maximum lactate dehydrogenase level was 1138 (691-2124) IU/L (Table 3).
Comment 6
coagulation level as optimal range, and maintain preload of Impella are recommended for 272
Response 6
We corrected the word.
Before 6
Conclusion
Be aware of hemolysis and maintain anticoagulation level as optimal range, and maintain preload of Impella are recommended for patients with lower PAPi.
After 6
Conclusion
Be aware of hemolysis and maintain anticoagulation level as optimal range, and maintain preload of Impella is recommended for patients with lower PAPi.

Reviewer 3 Report
The authors conducted a retrospective, single-centre study that considered 42 patients supported with Impella pump. They investigated the association of pulmonary artery pulsatility index (PAPi) with hemolysis during Impella support. The main findings of the study are: a PAPi cutoff of 1.3 correlates with hemolysis with high sensitivity and specificity. Lower PAPi (< 1.3) is associated with an OR of 10.69 for hemolysis.
The topic is of interest and the underlying hypothesis - RV dysfunction could be associated indirectly with hemolysis - is intriguing.
The present study, however, cannot be published in the present form, as it presents some methodological issues that have to be addressed by the authors.
Major revisions:
1) Definition of hemolysis: this is among the major weakness point. Indeed, plasma-free haemoglobin must be included as a marker of hemolysis, as its sensitivity and specificity allow precise identification of this adverse event. Many other explanations could sustain an increase of LDH and hemoglobinuria has a low sensitivity.
2) As recognized in the limitation section, the lack of RV function data and Impella flow represents the main obstacle to the reliability of the results.
Other comments:
1) The introduction is well written and clearly introduces the topic, with an appropriate choice of citations.
2) Methods: the indication to Impella support is vague. It is not arguable if patients were on cardiogenic shock. This information cannot be missing.
3) Patient selection: 3 patients did not receive Impella, therefore should not be considered excluded.
4) There are multiple references about low/high PAPi, but it is never defined precisely. PAPi cannot be categorized as low/high without any reference to the previously identified cutoff (RV dysfunction in RV-AMI and after LVAD).
6) Potential trigger of hemolysis, device upgrade and other clinical outcomes: these data are unclear and should be represented in a table or included in table 5.
7) Discussion: the main hypothesis (lines 243-249) has a pathophysiological rationale, but it should be supported by some references, i.e. studies on shear stress during Impella support etc.
Author Response
Reviewer #3
General comment
The authors conducted a retrospective, single-centre study that considered 42 patients supported with Impella pump. They investigated the association of pulmonary artery pulsatility index (PAPi) with hemolysis during Impella support. The main findings of the study are: a PAPi cutoff of 1.3 correlates with hemolysis with high sensitivity and specificity. Lower PAPi (< 1.3) is associated with an OR of 10.69 for hemolysis.
The topic is of interest and the underlying hypothesis - RV dysfunction could be associated indirectly with hemolysis - is intriguing.
The present study, however, cannot be published in the present form, as it presents some methodological issues that have to be addressed by the authors.
Response
We sincerely express our great appreciation for your comments that have improved our manuscript. According to the reviewer’s comment, we revised our manuscript. Please read through our updated manuscript.
Comment 1
Definition of hemolysis: this is among the major weakness point. Indeed, plasma-free haemoglobin must be included as a marker of hemolysis, as its sensitivity and specificity allow precise identification of this adverse event. Many other explanations could sustain an increase of LDH and hemoglobinuria has a low sensitivity.
Response 1
We sincerely agree with the reviewer’s critical comment. LDH elevation is also caused by other causes such as myocardial infarction, and might not be the best marker of hemolysis. However, we cannot measure the plasma free hemoglobin by the insurance and the measurement of plasma free hemoglobin is commercially not available in our country. We strengthened the limitation section.
Before 1
Limitations
We did not obtain data on plasma free hemoglobin levels and did not investigate the relationship between the occurrence of hemolysis and Impella support flow.
After 1
Limitations
We did not measure plasma free hemoglobin levels, which was one of the markers of hemolysis, because it was not covered by the insurance and cannot be measured commercially.
Comment 2
As recognized in the limitation section, the lack of RV function data and Impella flow represents the main obstacle to the reliability of the results.
Response 2
We sincerely agree with the reviewer’s critical comments. As mentioned in the limitation section, it is challenging to clearly depict the right ventricular function by transthoracic echocardiography and accurately measure TAPSE and/or RVFAC in patients with critical cardiogenic shock receiving many mechanical devices under limited physique. There are several papers which investigated the association between the degree of change of RV adaptation following axillary Impella insertion and early right ventricular failure following durable LVAD implantation, using parameters as follows; RAP, RAP/PCWP, and PAPi (J Heart Lung Transplant. 2021 Nov15;S1053-2498(21)02585-7). We did not include the data of RAP/PAWP due to large amount of missing data of PAWP.
In addition, the risk of developing RV dysfunction may vary by device type, although not investigated in detail thus far. There was a report that the impaired RV function caused by LMT-AMI cardiogenic shock, was partially restored during Impella CP support in adult pigs (Intensive Care Med Exp. 2020 12;8(1):41). On the other hand, the right ventricular failure following Impella insertion had also been reported (J Heart Lung Transplant. 2021 Nov15;S1053-2498(21)02585-7). Further investigation is needed to clarify the risk of developing RV dysfunction among Impella devices. It may be dependent on the underlying diseases. We strengthened the limitation section.
We did not investigate the relationship between the occurrence of hemolysis and Impella support flow precisely. However, univariable analysis for the hemolytic event in our study shown in Table 4 revealed no significant difference between Impella devices. We also showed as below the occurrence of hemolysis separately for each Impella device. There was no significant difference for the occurrence of hemolysis between Impella devices (p = 0.112), although Impella 5.0 had tendency of low risk for hemolysis. We added
|
  |
Impella 2.5 |
Impella CP |
Imeplla 5.0 |
|
Hemolysis |
11 (52%) |
7 (63%) |
2 (20%) |
|
Non-Hemolysis |
10 (48%) |
4 (36%) |
8 (80%) |
|
Total |
21 (50%) |
11 (26%) |
10 (24%) |
Before 2
Results
None
Limitation
We did not assess right ventricular function by using echocardiography. Accurate assessment of right ventricular function by echocardiography is often challenging due to the existence of multiple mechanical devices.
After 2
Results
Impact of PAPi on hemolysis:
During the Impella support, a total of 20 patients (48%) encountered hemolysis events; 11 patients (52%) in Impella 2.5, 7 patients (63%) in Impella CP, and 2 patients (20%) in Impella 5.0, which was not significantly different of incidence among Impella devices.
Limitation
We did not assess right ventricular function by using echocardiography. Accurate assessment of right ventricular function by echocardiography is often challenging due to the existence of multiple mechanical devices. We did not include the data of right atrial pressure/pulmonary artery wedge pressure due to large amount of missing data.
In addition, right ventricular function might change following Impella insertion (J Heart Lung Transplant. 2021 Nov15;S1053-2498(21)02585-7), and the incidence might vary depending on the device types.
Comment 3
The introduction is well written and clearly introduces the topic, with an appropriate choice of citations.
Response 3
We sincerely appreciate the reviewer’s comment.
Comment 4
Methods: the indication to Impella support is vague. It is not arguable if patients were on cardiogenic shock. This information cannot be missing.
Response 4
Thank you for the critical and important comment. Impella 2.5 and 5.0 had been approved for insurance since September 2017, and Impella CP had been commercially available since October 2019 in our country.
For that reason, we inserted Impella 2.5 and perform primary PCI in STEMI until October 2019, but inserted Impella CP after November 2019. If end-organ dysfunction still remained or progressed even after primary PCI, we added V-A CMO to maintain systemic perfusion.
In case of INERMACS Profile 1, i.e. refractory ventricular fibrillation, cardiac arrest on CRP, severe cardiogenic shock accompanied hepatorenal dysfunction such as SCAI-Shock stage E, we inserted V-A ECMO at first and perform CAG and PCI if needed. Because LV unloading with Impella during ECMO was associated with improved survival (Rev Cardiovasc Med. 2021 22;22(4):1503-1511), we added Impella 2.5 or CP for LV unloading in case of LV dysfunction and/or elevated LVEDP.
In case of INTERMACS Profile 2, i.e. acute decompensated heart failure of cardiomyopathy, we inserted Impella 5.0 for bridge to recovery or durable LVAD. We revised the description concerning device selection.
Before 4
Methods
Device selection:
The device type was selected according to the diameters of the access vessels and required support flow/duration. Veno-arterial extracorporeal membrane oxygenation (ECMO) was used concomitantly with Impella if support flow was not sufficient for end-organ recovery.
After 4
Methods
Device selection:
Impella was inserted via femoral or axillary artery. The device type was selected according to the diameters of the access vessels and the underlying disease, indications, and hemodynamics. All these decisions were made by the multidisciplinary team discussion.
For example, Impella 2.5/CP was inserted before primary percutaneous coronary intervention (PCI) in ST elevation myocardial infarction. We inserted Impella 2.5 until October 2019 and CP after November 2019, due to the issue of approved timing. If end-organ dysfunction persisted or rather progressed even after primary PCI, veno-arterial extracorporeal membrane oxygenation (ECMO) was used concomitantly with Impella for the end-organ recovery.
In case of INERMACS Profile 1, i.e. refractory ventricular fibrillation, cardiac arrest on cardiopulmonary arrest resuscitation, or severe cardiogenic shock accompanied by hepatorenal dysfunction, we inserted veno-arterial ECMO at first and performed PCI if needed. We inserted Impella 2.5 or CP in addition to veno-arterial ECMO in case of left ventricular dysfunction and/or elevated left ventricular end-diastolic pressure.
In case of INTERMACS Profile 2, i.e. acute decompensated heart failure of cardiomyopathy with progressive end-organ dysfunction against inotropes, we inserted Impella 5.0 for bridge to recovery or durable LVAD.
Comment 5
Patient selection: 3 patients did not receive Impella, therefore should not be considered excluded.
Response 5
We apologize for the misleading expressions. The three patients received Impella, however, they had surgical bleeding before Impella insertion and could not control bleeding or hemostat surgical bleeding after Impella insertion despite of low dose heparin purge solution, and eventually died due to blood loss. We revised the sentence.
Before 5
Method
Patient selection:
Of four patients, one patient had chest bleeding due to cardiopulmonary resuscitation-associated trauma and the other three patients had surgical bleeding before the insertion of Impella, who eventually died due to blood loss.
After 5
Method
Patient selection:
Of four patients, one patient had chest bleeding due to cardiopulmonary resuscitation-associated trauma and the other three patients had surgical bleeding, who eventually died due to blood loss, despite of low dose heparin purge solution.
Comment 6
There are multiple references about low/high PAPi, but it is never defined precisely. PAPi cannot be categorized as low/high without any reference to the previously identified cutoff (RV dysfunction in RV-AMI and after LVAD).
Response 6
Thank you for the critical comment. As suggested, there is no established cutoff to distinguish normal range of PAPI. In this study, we statistically calculated a cutoff that stratified the primary endpoint. We changed the term “low PAPi and normal PAPi” to “lower PAPi and higher PAPi”. In addition, we added the limitation section concerning PAPi never defined precisely thus far.
Before 6
See previous text
Limitation:
None
After 6
See updated text
Limitation:
Although, there are multiple references about PAPi, it is never defined precisely thus far. Further investigation is needed to clarify the association between the absolute value of PAPi and right ventricular function in patients with acute myocardial infarction and/or following LVAD implantation.
Comment 7
Potential trigger of hemolysis, device upgrade and other clinical outcomes: these data are unclear and should be represented in a table or included in table 5.
Response 7
Thank you for the important and critical comment. The patient who received Impella upgrade was only one and reached the VSR repair and achieved survival discharge. The other patients who received concomitant use of V-A ECMO were 10 patients. All these patients were lower PAPi (<1.3) group. We showed the clinical outcomes these patients as below. We also added the proportion of device upgrade/concomitant use of ECMO into the Table.
|
  |
Impella upgrade (n=1) |
Concomitant use of ECMO (n=10) |
|
Clinical outcomes |
||
|
Suction events |
1 |
4 (40%) |
|
Pump thrombosis |
0 |
1 (10%) |
|
Hemolysis |
1 |
6 (60%) |
|
Recovery without mechanical circulatory support |
1 |
8 (80%) |
|
Durable left ventricular assist device implantation |
0 |
0 (0%) |
|
30-day survival |
1 |
7 (70%) |
|
Survival discharge |
1 |
4 (40%) |
Before 7
Table
See previous Table 5
Results
Other clinical outcomes:
None.
After 7
Table
See updated Table 6
Results
Other clinical outcomes:
The prevalence of clinical outcomes other than hemolysis were stratified by PAPi (Table 6). There was significant higher rate of Impella upgrade and/or concomitant use of ECMO in the lower PAPi group.
Comment 8
Discussion: the main hypothesis (lines 243-249) has a pathophysiological rationale, but it should be supported by some references, i.e. studies on shear stress during Impella support etc
Response 8
We sincerely appreciate and agree with the reviewer’s comment. Impella related hemolysis occur by increased shear stress. We revised the discussion section.
Before 8
Discussion
PAPi and hemolysis:
In addition to the above-described risk factors of hemolysis, we would like to propose a recently-introduced hemodynamic parameter: PAPi [9]. A decrease in preload on the left ventricle due to impaired right ventricular function, indicated by lower PAPi, would reduce intra-device blood stream as well as promote micro-thrombus formation in left ventricle and micro-axial pump, both of which might increase shear stress and facilitate hemolysis.
After 8
Discussion
PAPi and hemolysis:
In addition to the above-described risk factors of hemolysis, we would like to propose a recently-introduced hemodynamic parameter: PAPi [9]. A decrease in preload on the left ventricle due to impaired right ventricular function, indicated by lower PAPi, would reduce intra-device blood flow in left ventricle and might increase shear stress and facilitate hemolysis.

Round 2
Reviewer 3 Report
After the review, the manuscript is significantly improved.
Many limitations persist, but they are clearly stated in the limitation section.
Therefore, I think that this manuscript is worth publishing due to its novel and interesting hypothesis.